# Bioactive Compounds of Underground Valerian Extracts and Their Effect on Inhibiting Metabolic Syndrome-Related Enzymes Activities

**DOI:** 10.3390/foods12030636

**Published:** 2023-02-02

**Authors:** Cheng-Rong Wu, Shih-Yu Lee, Chien-Hung Chen, Sheng-Dun Lin

**Affiliations:** 1Department of Food Science and Technology, Hungkuang University, Taichung 433304, Taiwan; 2College of Nursing, Hungkuang University, Taichung 433304, Taiwan; 3Byrdine F. Lewis College of Nursing and Health Professions, Georgia State University, Atlanta, GA 30302, USA; 4Mei Toong Co., Ltd., Nantou 545, Taiwan

**Keywords:** *Valeriana officinalis*, bioactive compounds, antioxidants, metabolic syndrome, enzymes

## Abstract

Extractions of the underground parts of valerian were prepared with water and ethanol (25–95%) at 25–75 °C. Extraction yields, bioactive compounds, and the 1,1-Diphenyl-2-picrylhydrazyl (DPPH) radical scavenging ability of lyophilized extracts were determined. The inhibitory effects of the extracts, valerenic acid derivatives and phenolic acids, on metabolic syndrome (MS)-related enzymes activities were further examined. Both roots and rhizomes extracted with 95% ethanol at 75 °C had the highest levels of bioactive compounds. The antioxidant capacity and inhibition of MS-related enzymes of the roots extract were better than those of the rhizomes. The roots extract more strongly inhibited pancreatic lipase (inhibition of 50% of enzyme activity (IC_50_), 17.59 mg/mL), angiotensin-converting enzyme (ACE, IC_50_, 3.75 mg/mL), α-amylase (IC_50_, 12.53 mg/mL), and α-glucosidase (IC_50_, 15.40 mg/mL). These four phenolic acids inhibited the activity of MS-related enzymes. Valerenic acid demonstrated more of an inhibitory ability for ACE (IC_50_, 0.225 mg/mL, except for caffeic acid) and α-glucosidase (IC_50_, 0.617 mg/mL) than phenolic acids. Valerian extract inhibited key enzyme activities that were associated with obesity (lipase), hypertension (ACE), and type 2 diabetes (α-amylase and α-glucosidase), suggesting that it is a potential candidate for the development of functional supplements.

## 1. Introduction

*Valeriana officinalis* (valerian) is a flowering plant of the *Caprifoliaceae* family that is commonly used to relieve sleep and anxiety disorders. It is safe for short-term use, but its side effects include headache, stomach upset, etc. [1,2]. Valerian is rich in phytochemicals, such as phenolic compounds, which are related to antioxidant activity. More than 150 different phytochemical compounds have been identified in valerian root, with the main components being sesquiterpenes (e.g., valerenic acid), iridoids, and flavonoids [3]. Among these components, valerenic acid exerts anxiolytic activity by regulating the γ-aminobutyric acid (GABA) receptor, and it is often utilized as a medicinal quality indicator [4,5,6].

Phenolic compounds inhibit free radicals to link with antioxidant activity, limiting nutrient oxidation by inhibiting oxidative chain reactions, which positively affect cardiovascular health and have anti-inflammatory, antioxidant, and anti-cancer properties [7]. Clinicians highly value phenolic compounds in food, and research has confirmed that plant extracts that are rich in phenolic compounds (i.e., phenolic acids) inhibit pancreatic lipase, α-amylase, α-glucosidase, and angiotensin-converting enzyme (ACE) activity [8,9]. Moreover, studies have revealed that valerian roots extract has antioxidant properties, slowing cell damage and aging [10,11]. 

Theoretically, valerian could be utilized to treat metabolic syndrome (MS); however, to date, few studies [3,12,13] have investigated phenolic compounds and the enzymatic activities that are related to the inhibition of MS. In a mouse model [12], it was demonstrated that above-ground valerian extracts inhibited the differentiation of lipid content in 3T3-L1, a cell line used to study the fundamental cellular mechanisms related to MS. Another study [13] used 70% ethanolic valerian root extract and demonstrated that iridoids enhance autophagy to break down lipid droplets, thereby relieving fatty liver.

MS is a group of syndromes that increase the risk of coronary heart disease, diabetes, stroke, and other serious health problems [14]. At least one-quarter of the world’s population have MS, with massive annual healthcare costs, causing a loss of productivity in the workforce [15]. A healthy lifestyle to maintain an ideal body weight could prevent MS, and medications are available to treat MS; however, medicines have side effects, which also increases healthcare costs. Therefore, finding natural substances that can modulate MS is important. Systematic review and meta-analysis revealed that green tea, phaseolus vulgaris, garcinia cambogia, nigella sativa, and ferulic acid have an effect on metabolism syndrome-related enzymes [16,17,18]; among them, ferulic acid [18] and garcinia family [16] are phenolic acids. This study investigated the extraction yield and bioactive compounds of lyophilized extracts from the underground parts (roots and rhizomes) of *V. officinalis.* Further, the inhibitory effects of the extracts on the specific MS-related enzymes activities were examined: pancreatic lipase, ACE, α-amylase, and α-glucosidase.

## 2. Materials and Methods

### 2.1. Plant Materials

Dried underground valerian (*V. officinalis*) parts, including the roots and rhizomes (Figure 1), were harvested from Guizhou, China (longitude 108.83967, latitude 27.69956). Valerian was donated by KO DA Pharmaceutical Co. Ltd. (Taoyran, Taiwan, China). Foreign matter was removed using a 24-mesh sieve. The valerian samples were divided into four groups of 600 g each, and the roots and rhizomes were separated with scissors, milled, and sieved (<0.7 mm). Then, the samples were packed in laminated bags (PET/Al/LLDPE) and stored in a −25 °C freezer.

### 2.2. General Experimental Procedure

Valerian powder (15 g) was added to 150 mL of water or various concentrations (25%, 50%, 75%, and 95%) of ethanol in a shaking water bath at 150 rpm for 30 min at 25–75 °C, then centrifuged at 6932× *g* (10,000 rpm, 20 min), and filtered through Advantec No.1 filter paper. The residue was re-extracted with 150 mL portions of the solvent, as described above. The combined filtrate was concentrated under reduced pressure at a 50 °C water bath and freeze-dried. The lyophilized extracts obtained by extracting the samples with water, and 25%, 50%, 75%, and 95% ethanol in a 25 °C water bath were labeled 25C0E, 25C25E, 25C50E, 25C75E, and 25C95E, respectively. The first set of numbers and letters indicates the bath temperature in Celsius, and the second set of numbers and letters indicates the percentage of ethanol. All treatments were randomly generated and performed in quadruplicate.

### 2.3. Chemical Materials

Methanol (purity > 99.9%), acetonitrile (purity > 99.9%), and phosphoric acid (86.2%) were purchased from Avantor Performance Material (Center Valley, PA, USA). Valerenic acid (purity > 99.9%) and hydroxyvalerenic acid (purity > 99.9%) were purchased from ChromaDex (Santa Ana, CA, USA). Chlorogenic acid (purity > 95%), caffeic acid (purity > 97%), protocatechuic acid (purity > 97%) and gallic acid (purity > 98%), Folin–Ciocalteu’s phenol reagent (2N), DPPH (purity > 95%), DMSO (purity > 99.5%), sodium dodecyl sulfate (purity > 97%), starch (purity > 99.9%), acarbose (purity > 95%), ascorbic acid (purity > 99%), α-tocopherol (purity > 96%), BHA (purity > 98.5%), 3,5-dinitrosalicylic acid (purity > 98%), porcine pancreatic α-amylase (EC 3.2.1.1), p-nitrophenyl-α-D-glucopyranoside (purity > 98%), p-nitrophenyl laurate (purity > 98%), α-glucosidase (EC 3.2.1.20), rabbit lung ACE (EC 3.4.15.1), hippuryl-l-histidyl-l-leucine (purity > 98%), hippuric acid (purity > 98%), captopril (purity > 99.5%), Orlistat (purity > 98%), acetoxyvalerenic acid (purity > 95%), and Triton X-100 were purchased from Sigma-Aldrich (St. Louis, MO, USA). Sodium chloride (purity > 98.8%) was purchased from Shimakyu’s Pure Chemicals (Osaka, Japan). Anhydrous sodium carbonate (purity > 99%), sodium dihydrogen phosphate (purity > 98%), sodium phosphate dibasic dodecahydrate (purity > 98%), and hydrochloric acid (37%) were purchased from Union Chemical Work (Hsinchu, Taiwan). Formic acid (purity > 98%) was purchased from Honeywell International (Lower Saxnoy, Germany). Potassium sodium tartrate tetrahydrate (purity > 99.5%) was purchased from Wako Pure Chemical (Osaka, Japan). Tris(hydroxymethyl) aminomethane (purity > 98%) was purchased from Amresco (Solon, OH, USA). Ethanol (95%) was purchased from Taiwan Tobacco & Liquor (Tainan, Taiwan).

### 2.4. Determination of Extraction Yields and Bioactive Compounds

The extraction yield of all treatments is calculated as (freeze-dried extract weight/sample weight) × 100%. Total phenols levels of the extracts were analyzed following the method described by Mau et al. [19]. The total phenols content was calculated based on the calibration curve of gallic acid (absorbance at 760 nm = 0.0008 C_gallic acid_ (μg/mL) + 0.0077, R² = 0.9993). The results were expressed as milligrams of gallic acid equivalent (GAE) per gram of lyophilized extract. The phenolic acid levels in the lyophilized extract were determined based on the method of Sarikurkcu et al. [20], with some modifications. Each extract (100 mg) was dissolved in methanol (4 mL) using an ultrasonic bath at 53 kHz for 15 min, and then the volume was adjusted to 5 mL with methanol, and centrifuged at 1900× *g* for 5 min. The solution was filtered with a nylon syringe filter (13 mm × 0.45 μm) before injection into an HPLC system consisting of a Hitachi 5110 pump, a Hitachi 5430 Diode array detector (Tokyo, Japan), and a Phenomenex Luna C18 (2) column (250 mm × 4.6 mm, 5 μm particle size; Phenomenex, Torrance, CA, USA). The mobile phase was composed of 0.1% (*v*/*v*) formic acid (solvent A) and methanol (solvent B) at a flow rate of 0.4 mL/min. Elution was carried out using a linear gradient as follows: 2% B from 0 min to 6 min, 2–25% B from 6 min to 12 min, 25–50% B from 12 min to 20 min, 50–95% B from 20 min to 28 min, 95% B from 28 min to 34 min, 95–2% B from 34 min to 35 min, and 2% B from 35 min to 45 min. The sample injection volume was 10 μL. Gallic acid and protocatechuic acid were detected at a wavelength of 280 nm. Chlorogenic, caffeic, and rosmarinic acids were detected at 330 nm.

Valerenic acid derivatives in the extracts were determined using a modified method by Donovan et al. [21]. The sample preparation method and HPLC system were the same as for the analysis of phenolic acids. The mobile phase composed of acetonitrile (solvent A) and 0.1% phosphoric acid (solvent B) at a flow rate of 1.5 mL/min, and UV detection was at 218 nm. Elution was carried out using a linear gradient as follows: 60–20% B from 0 min to 20 min, and 60–20% B from 20 min to 25 min. The sample injection volume was 10 μL. The valerenic acid derivative levels of the extract were calculated based on the calibration curves of hydroxyvalerenic, acetoxyvalerenic, and valerenic acids. Each analysis was carried out in quadruplicate.

### 2.5. Antioxidant Properties

The scavenging ability of each extract on DPPH radicals was determined based on the method by Shimada et al. [22]. Each extract was prepared in methanol at 0–1000 μg extract/mL, and 4 mL was mixed with 1 mL of methanolic solution containing DPPH radicals to a final concentration of 0.2 mM DPPH. After the mixture was shaken vigorously, it was left to stand in the dark for 30 min, and then the absorbance of the mixture was measured at 517 nm against a blank. The EC_50_ value (mg extract/mL) is the effective concentration at which the DPPH radicals were scavenged by 50%. Ascorbic acid, BHA, α-tocopherol, acetoxyvalerenic acid, and valerenic acid were compared. Each analysis was carried out in quadruplicate.

### 2.6. Lipase, ACE, α-Amylase, and α-Glucosidase Inhibition Assays

The inhibitory abilities of the extracts, phenolic acids, or valerenic acid derivatives; or orlistat, acarbose, or captopril solutions on lipase, α-amylase, α-glucosidase, and ACE were determined according to a previous study [23,24]. The lipase, ACE, α-amylase, and α-glucosidase inhibitory activities were expressed as the percentage of inhibition, and IC_50_ value (mg/mL) is the inhibition concentration at which enzyme activity was inhibited by 50%. Each analysis was carried out in quadruplicate. The analytical method is briefly described below. In the lipase inhibitory activity, 25 μL of extracts (roots 0–60 mg/mL and rhizomes 0–100 mg/mL), orlistat (0–125 ng/mL), phenolic acids (gallic, protocatechuic, and caffeic acids at 0–1.25 mg/mL; and chlorogenic acid at 0–2.5 mg/mL), valerenic acid derivatives (0–2 mg/mL) dissolved in 5% DMSO solution, were incubated with 50 μL substrate and 25 μL lipase (150 U/mL) for 30 min at 37 °C. The supernatant of the reacted solutions was read at 405 nm using an ELISA reader. Orlistat was used as a positive control. 

In the ACE-inhibitory activity, a mixture of 100 μL of ACE solution (2.5 mU/mL), 100 μL of extract (0–20 mg/mL), captopril (0–3.33 ng/mL), phenolic acids (gallic and protocatechuic acid at 0–3.33 mg/mL, chlorogenic acid at 0–6.67 mg/mL, and caffeic acid at 0–0.333 mg/mL), or valerenic acid derivatives (0–1 mg/mL) solution was preincubated for 30 min at 37 °C. The above mixture was added with 100 μL of 3 mM hippuryl-L-histidyl-L-leucine (1 mM final concentration), and the mixture was incubated for 30 min at 37 °C. The reaction was stopped by adding 100 μL of 12% phosphoric acid. The hippuric acid was determined via HPLC. Captopril was used as a positive control. 

In the α-amylase inhibitory activity, a total of 200 μL of extract (0–50 mg/mL), acarbose (0–25 μg/mL), phenolic acids (0–2.5 mg/mL), or valerenic acid derivative (0–2 mg/mL) solutions; and 200 μL of 0.02 M sodium phosphate buffer (pH 6.9 with 0.006 M NaCl) containing α-amylase solution (10 U/mL) were incubated in a shaking bath (37 °C) at 100 rpm for 45 min. Then, 400 μL of a 0.5% starch solution was added to each tube and incubated in a shaking bath (37 °C) for 10 min. The reaction was stopped with 1.0 mL of dinitrosalicylic acid color reagent. The test tubes were incubated in a boiling water bath for 10 min and cooled to room temperature. The reaction mixture was then diluted after adding 3 mL of distilled water, and the absorbance was measured at 540 nm using a spectrophotometer. The readings were compared with the controls containing buffer instead of the sample extract. Acarbose was used as a positive control. 

In the α-glucosidase inhibitory activity, 100 μL of extract (0–30 mg/mL), acarbose (0–400 μg/mL), phenolic acids (0–6 mg/mL), or valerenic acid derivatives (0–2 mg/mL) solution, and 100 μL of α-glucosidase solution (1 U/mL) in a 0.1 M phosphate buffer (pH 6.9) was incubated at 25 °C for 10 min. Then, 50 μL of 5 mM p-nitrophenyl-α-d-glucopyranoside solution was added to the 0.1 M phosphate buffer (pH 6.9). The mixture was incubated at 25 °C for 5 min, followed by the addition of 25 μL of 0.1 M sodium carbonate solution to stop the reaction. The absorbance was measured at 405 nm using an ELISA reader. Acarbose was used as a positive control.

### 2.7. Statistical Analysis

All measurements were performed in quadruplicate. The data were subjected to analysis of variance with SAS software (SAS Institute, Cary, NC, USA). When a significant difference was found among the treatment groups, Duncan’s multiple range tests were performed to determine the differences among the mean values at a level of α = 0.05.

## 3. Results and Discussion

### 3.1. Extract Yield and Bioactive Compounds

The extraction was performed using solvents (i.e., water and food-grade ethanol) to separate the desired natural products from the raw materials [25]. In this study, the extraction yield of valerian extract products decreased with increasing ethanol concentration, but increased with increasing extraction temperature (Table 1). 

The total phenols extracted from the roots and rhizomes are detailed in Table 1. Using 95% ethanol in a water bath at 75 °C, the highest amount of total phenols from roots (R75C95E) and rhizomes (Rh75C95E) were extracted at 33.16 and 26.78 mg GAE/g lyophilized extract, respectively. Considering the extraction yield, the total phenols content of the powder sample was used to represent the extraction efficiency. Roots and rhizomes were extracted with 50% and 75% ethanol in a water bath at 75 °C, and the highest levels of total phenols content were 7.56 and 5.55 mg GAE /g powder, respectively (Table 1). The total phenols content extracted was higher than that in previous studies [26,27], indicating that the solvent polarity and heating methods used in this study are the best extraction method. We also determined the phenolic acids content of roots and rhizomes extracts (Table 2). The HPLC profiles of phenolic acids standards, valerian roots, and rhizomes extracts are shown in Appendix A. In the 75 °C water bath, the contents of four phenolic acids, including gallic, protocatechuic, chlorogenic, and caffeic acids increased with increased ethanol concentration, indicating the solubility of phenolic acids in different solvents [28]. In both the roots and rhizomes extracts, the phenolic acid compounds of 75C95E were mainly protocatechuic acid. The HPLC profiles of the valerenic acid derivatives standards, valerian roots, and rhizomes extracts are shown in Appendix A. The valerenic acid derivatives extracted from roots and rhizomes are detailed in Table 3. The levels of three valerenic acid derivatives increased with increasing ethanol concentration and water bath temperature, which is in line with a previous study [29]. Thus, 75C95E had the highest valerenic acid derivatives, and the level in rhizomes was higher than that in roots. In rhizomes, acetoxyvaleric acid had the highest content (24.2 mg/g) of valeric acid derivatives, accounting for 72.85% of the total valeric acid derivatives. 

### 3.2. Antioxidant Properties

Human disease (such as diabetes mellitus and atherosclerosis) progression is linked to free radicals [30]. Naturally occurring antioxidants retard the progress of many chronic human diseases by scavenging free radicals. The ability of extracts to scavenge DPPH radicals displayed a dose-dependent effect (Figure 2). In comparison with rhizomes, the roots extract had a better antioxidant property. When the concentration of the roots extract was 0.2 mg extract/mL, 75C95E had the highest DPPH radical scavenging activity (26.65%); the concentration increased to 1.0 mg extract/mL and the DPPH radical scavenging activity increased to 86.83%. The EC_50_ values (mg extract/mL) of roots extracts for scavenging DPPH free radicals were 0.352 (75C95E), 0.374 (75C75E), 0.535 (75C50E), 0.639 (75C25E), and 0.708 (75C0E) (Table 4). Thus, the higher ethanol concentration as the extraction solvent, the stronger the ability of the obtained extract to scavenge DPPH radicals, and the lower the EC_50_ value. The EC_50_ value of the antioxidant activity of the valerian roots essential oil was 493.40 μg/mL [31]. In another study on valerian roots extracted with methanol as the extraction solvent, the 50% DPPH radicals scavenging concentration of the extract obtained with ultrasonic-assisted extraction was 54.6 μg/mL [10], which was related to the phenolic compounds content in the extract [10,32].

Roots extracts had better antioxidant properties, which may be attributed to the fact that the levels of phenolic compound in the roots extracts were higher than those in the rhizomes extracts (Table 2 and Table 3). Valerenic and acetoxyvaleric acid (0–250 μg/mL) were not able to scavenge the DPPH radicals in this study. The EC_50_ values (µg/mL) of ascorbic acid, BHA, and α-tocopherol in scavenging DPPH radicals were 15.67, 16.48, and 16.09, respectively. Valerian extracts were less effective than ascorbic acid, BHA, and α-tocopherol at DPPH scavenging, but these standards are additives that are used in foods or that are present at milligram levels. However, valerian extract can be used as a food supplement at the gram level.

### 3.3. Inhibitory Effects on Lipase

Pancreatic lipase is a key enzyme that regulates lipid absorption [33]. Pancreatic lipase inhibition by the extracts was dose-dependent (Figure 3A). At the same dosage, the roots extract had a better inhibitory ability than that of the rhizomes extract. When the dosage of the roots extract was 10 mg extract/mL, 75C95E had the highest inhibitory ability of pancreatic lipase (45.54%); the dosage increased to 60 mg extract/mL and the inhibition of pancreatic lipase increased to 86.00%. The IC_50_ values (mg extract/mL) for pancreatic lipase were 50.75 (0E), 41.84 (25E), 38.28 (50E), 30.67 (75E), and 17.59 (95E) (Table 5). The inhibitory effect of the extract with 95% ethanol was the best.

Orlistat is an anti-obesity agent that potently inhibits pancreatic lipase [34], and it also improves glycemic control in type 2 diabetes [35]. However, it may increase the risk of colon cancer [36] and reduce the absorption of the fat-soluble vitamins A, D, E, and K [35]. With increasing orlistat concentration, the inhibition of pancreatic lipase was better (Figure 3A). When the concentration was 9.648 ng/mL, IC_50_ was achieved (Table 6), which was much lower than the IC_50_ values of extracts.

When the concentrations of four phenolic acids (gallic, protocatechuic, chlorogenic, and caffeic acid) increased, a more potent inhibition of pancreatic lipase was observed (Figure 4A). The IC_50_ values of gallic, protocatechuic, chlorogenic, and caffeic acid for inhibiting pancreatic lipase were 0.623, 0.673, 1.108, and 0.726 mg/mL, respectively (Table 6). Thus, gallic acid had the best inhibitory ability among the four phenolic acids on lipase activity. Neither the standardized samples of valerenic acid nor acetoxyvalerenic acid effectively inhibited pancreatic lipase activity at 0.5–2 mg/mL, which indirectly indicates that the inhibition of pancreatic lipase activity is related to phenolic acids. Other studies support that phenolic compounds in vegetable and fruit extracts inhibit pancreatic lipase activity [8,37].

### 3.4. Inhibitory Effects on ACE

ACE is a carboxypeptidase, which plays an important role in hypertension management and cardiovascular protection. Thus, ACE inhibition may be a useful treatment for patients with MS [38]. The ACE inhibition by the extract was concentration-dependent (Figure 3B). The roots extract had a better inhibitory ability on ACE than that of the rhizomes extract at the same dosage. When the concentration was 2.5 mg extract/mL, 75C95E had the highest ACE inhibitory abilities (39.72%); the dosage increased to 20 mg extract/mL, and ACE inhibition was 77.31%. The IC_50_ value (mg extract/mL) of the inhibition of ACE activity by the roots extracts (Table 5) shows that their inhibitory abilities were 75C95E (3.75), 75C75E (4.02), 75C50E (4.24), 75C25E (11.22), and 75C0E (13.66).

At a captopril concentration of 0.33 ng/mL, the inhibition of ACE reached 40%. As the concentration increased, its inhibitory ability against ACE also increased (Figure 3B). The IC_50_ value of captopril on ACE inhibition was 0.498 ng/mL (Table 6), which was much lower than those of valerian extracts (Table 5). As the concentrations of valerenic, gallic, protocatechuic, chlorogenic, and caffeic acid increased, the inhibition of ACE increased (Figure 4B). The IC_50_ values of valerenic, gallic, protocatechuic, chlorogenic, and caffeic acid for inhibiting ACE activity were 0.225, 2.100, 2.462, 4.803, and 0.094 mg/mL, respectively (Table 6). Caffeic acid demonstrated the most potent inhibition of ACE, followed by valeric, gallic, protocatechuic, and chlorogenic acid. The dosage of acetoxyvaleric acid was developed at 2 mg/mL, and its inhibitory ability on ACE was only 15%, which indirectly confirms that the ability of extracts to inhibit ACE is mainly related to the valerenic acid and phenolic acids contents. Captopril demonstrated the strongest ACE inhibition, but it has side effects such as angioedema [39].

Both the roots and rhizomes extracts inhibited ACE activity, especially 95% ethanol extract. Notably, although the rhizomes extracts prepared with the same extraction solvent contained higher concentrations of valerenic acid than the root extracts, their ability to inhibit ACE showed the reverse, which may be related to the total phenols content of roots extracts being higher than that of rhizomes extracts (Table 2 and Table 3). Many studies have confirmed that plant phenolic compounds are the main factors inhibiting ACE. For example, these include *Echinacea* flower [23], oolong tea [40], and asparagus hard-stem [41]. 

### 3.5. Inhibitory Effects on α-Amylase

Type 2 diabetes treatment can decrease postprandial hyperglycemia by inhibiting carbohydrate hydrolases in the gastrointestinal tract [42]. Compared with the same dosage, the inhibitory ability of roots extract on α-amylase was better than that of rhizomes extract. When the roots extracts were prepared at 10–50 mg extract/mL, the ability to inhibit α-amylase activity increased with an increasing extract concentration in a dose-dependent manner (Figure 3C). At low concentrations (10 mg extract/mL), 75C95E had the highest degree of inhibition on α-amylase, at 45.16%. At a high concentration (50 mg extract/mL), the inhibitory ability of roots extracts on α-amylase was 93.80%. As shown in Table 5, the IC_50_ value (mg extract/mL) of the inhibition of roots extracts on α-amylase was 75C95E (12.53), 75C75E (20.65), 75C50E (28.23), 75C25E (30.41), and 75C0E (32.32). The R75C95E demonstrated the most strongly inhibitory ability of α-amylase. Compounds that inhibit α-amylase activity in methanol extracts from the upper part of *V. officinalis* may result from phenolic compounds [17]. Other studies have confirmed that plant phenolic compounds inhibit α-amylase activity. For example, phenolic compounds in the ethanol–water extract of *Citrus aurantium* (L) peel strongly inhibit α-amylase activity; at 1.0 mg extract/mL, α-amylase activity is inhibited by up to 76% [43].

At 5–25 μg/mL acarbose, the inhibitory ability of α-amylase was 47–88% (Figure 3C), and its IC_50_ value was 5.40 μg/mL (Table 6). The main phenolic acids in valerian extracts were analyzed for their ability to inhibit α-amylase activity. When the dosage was 0–2.5 mg/mL, the inhibition ability of gallic, protocatechuic, chlorogenic, and caffeic acid on α-amylase were 0–95.22%, 0–88.03%, 0–89.52%, and 0–89.35% (Figure 4C). Gallic acid had the lowest IC_50_ value (1.258 mg/mL) for inhibiting α-amylase (Table 6). Valeric acid and acetoxyvaleric acid (0.5–2 mg/mL) could not effectively inhibit α-amylase activity in this study, which indirectly indicates that the ability of the extract in this study to inhibit α-amylase was primarily derived from phenolic compounds. 

These extracts obtained using 95% ethanol have the best inhibitory effect on α-amylase, and the inhibitory ability of 75C95E on α-amylase activity in roots (IC_50_, 12.53 mg extract/mL) is higher than that in rhizomes (IC_50_, 21.99 mg extract/mL). The total phenols (33.16 mg GAE/g lyophilized extract) and total contents of four phenolic acids (3.08 mg/g lyophilized extract) in roots R75C95E were significantly higher than those in rhizomes 75C95E (Table 2 and Table 3). The findings indirectly confirmed that the abilities of roots and rhizomes extracts to inhibit α-amylase activity were related to the content of phenolic acids, which is in line with prior in vitro studies [20,23].

Acarbose is commonly used as an α-amylase and α-glucosidase inhibitor for type-2 diabetes. This study showed that acarbose was more effective in inhibiting α-amylase activity than valerian extract, valeric acid derivatives, and phenolic acids. Nevertheless, it has side effects and leads to the abnormal bacterial fermentation of undigested carbohydrates in the colon [44].

### 3.6. Inhibitory Effects on α-Glucosidase

α-Glucosidase breaks down disaccharides and oligosaccharides, and when it is inhibited, it can reduce postprandial blood glucose [45]. In this study, the inhibition of α-glucosidase by extracts increased as the concentration increased (Figure 3D). The inhibitory ability of roots extract on α-glucosidase was better than that of rhizomes extract. When the concentration of the roots extract was 5 mg extract/mL, 75C95E had the highest inhibitory ability of α-glucosidase (31.71%); the dosage increased to 30 mg extract/mL and the inhibition of α-glucosidase increased to 72.85%. The extracts from 95% ethanol extraction demonstrated the best inhibitory ability of α-glucosidase, while the water extract showed the lowest. Valerian extract may delay the digestion and absorption of carbohydrates, thereby inhibiting postprandial hyperglycemia.

As the concentration of valerenic acid derivatives and phenolic acids increased, their ability to inhibit α-glucosidase activity was also better (Figure 4D). The IC_50_ values (mg/mL) of valerenic, acetoxyvalerenic, gallic, protocatechuic, chlorogenic, and caffeic acid on α-glucosidase inhibition was 0.617, 1.827, 2.164, 3.721, 5.524, and 1.289, respectively (Table 6). Valerenic acid demonstrated the best inhibition of the activity of α-glucosidase. 

Based on the IC_50_ data in Table 5, 75C95E extracts at 17.59 and 40.92 mg/mL for roots and rhizomes, respectively, exhibited a comparable lipase inhibition effect to orlistat at 9.648 ng/L (Table 6). ACE inhibition by R75C95E and Rh75C95E was comparable to that of captopril (IC_50_ = 0.498 ng/mL) when the added doses were increased to 3.75 and 4.87 mg/mL, respectively (50% inhibition). Additionally, at doses of 12.53 mg/mL (R75C95E) and 21.99 mg/mL (Rh75C95E), these extracts demonstrate an equivalent level of inhibition against α-amylase, compared to that of acarbose applied at a dose of 5.404 μg/mL (50% inhibition). For α-glucosidase, the estimated IC_50_ values for R75C95E and Rh75C95E were 15.40 and 17.10 mg/mL, respectively, while the acarbose dose was 95.47 μg/mL. Therefore, theoretically, a combination of 49.27 mg R75C95E or 84.88 mg Rh75C95E may result in a 50% inhibition of lipase, ACE, α-amylase, and α-glucosidase activities. To increase the inhibition of pancreatic lipase, ACE, α-amylase, and α-glucosidase to 70%, the dosage of 75C95E in roots and rhizome extracts should be 104.45 and 141.52 mg extract/mL (Figure 3). The 75C95E extracts prepared from roots and rhizomes may induce delayed lipid and carbohydrate digestion, thereby inhibiting dyslipidemia, hypertension, and hyperglycemia.

## 4. Conclusions

With the rapid development of food processing technologies, researchers have been increasingly interested in developing natural products as dietary supplements to meet consumers’ health needs. Valerian extracts prepared with 95% ethanol as the extraction solvent and in a 75 °C water bath produced the highest bioactive component levels. The advantage of the specific extraction method is that ethanol is non-toxic, easy to mix with water, and low-cost; however, disadvantages exist since ethanol is flammable. There are other possible extraction methods, such as ultrasonic-assisted extraction, which is low-cost and high-yield; however, special equipment is required. The total phenols content of the roots extract was higher than that of the rhizomes. The phenolic acids in the roots and rhizomes mainly consisted of protocatechuic acid, but the total valeric acid content of the rhizomes was higher than that of the roots. The inhibition of R75C95E on pancreatic lipase, α-amylase, α-glucosidase, and ACE was higher than that of Rh75C95E. It is important to pay attention to these specific extracts as they have the highest inhibitory effects on MS-related enzymes activities. In conclusion, valerian roots and rhizomes extracts can be considered as a source for functional food development. Additional in vivo studies are required to investigate valerian extract, for preventing and managing MS.

## Figures and Tables

**Figure 1 foods-12-00636-f001:**
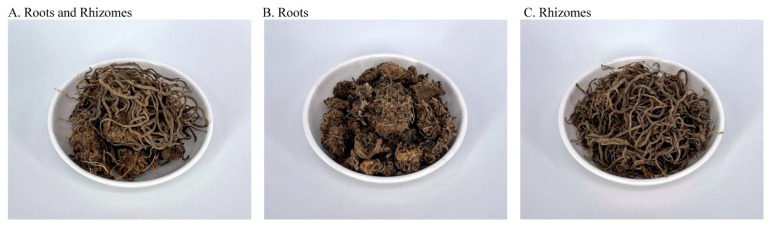
The underground parts of dried valerian.

**Figure 2 foods-12-00636-f002:**
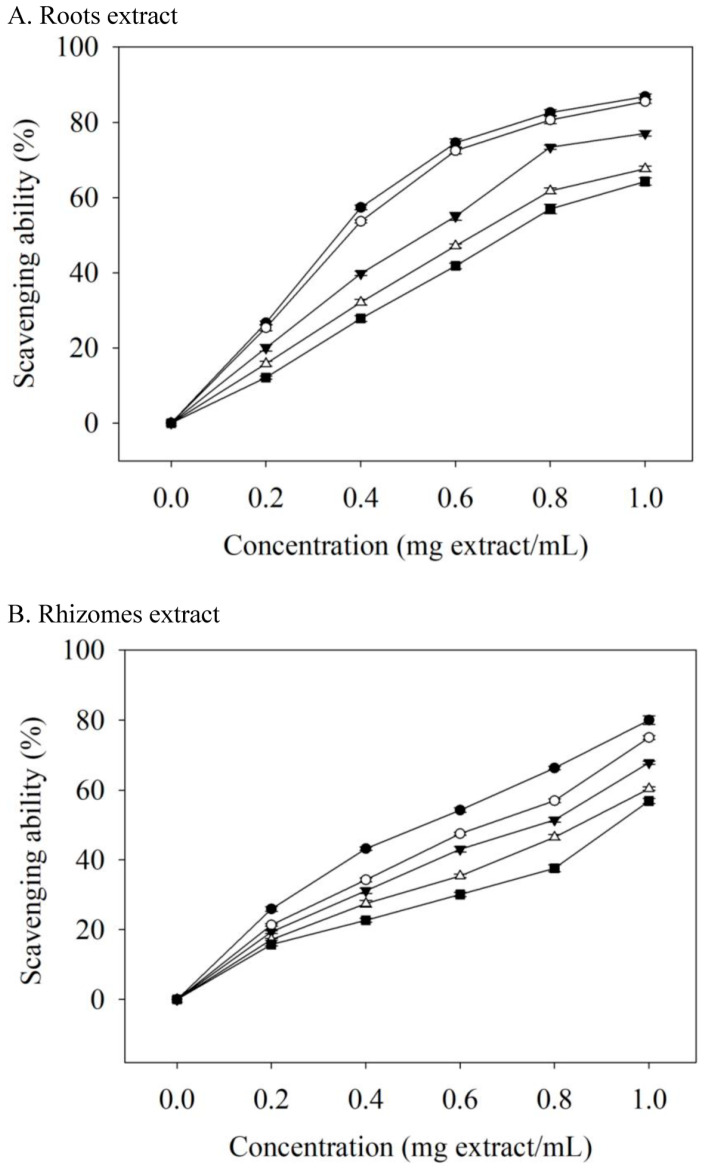
Scavenging ability on DPPH radicals of valerian extracts. Each value is expressed as the mean ± standard deviation (*n* = 4). 75C0E (■), 75C25E (△), 75C50E (▼), 75C75E (○), 75C95E (●).

**Figure 3 foods-12-00636-f003:**
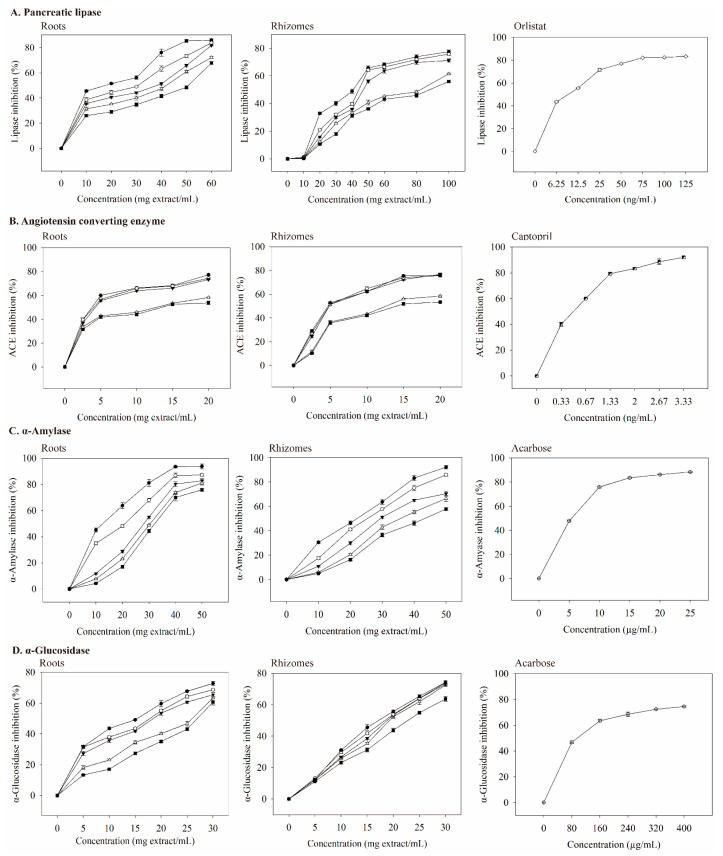
Inhibitory effects of valerian extracts, orlistat, captopril, and acarbose against enzymes linked with metabolic syndrome. Each value is expressed as mean ± standard deviation (*n* = 4). 75C0E (■), 75C25E (△), 75C50E (▼), 75C75E (○), 75C95E (●), orlistat (◇), captopril (◩), and acarbose (⊕).

**Figure 4 foods-12-00636-f004:**
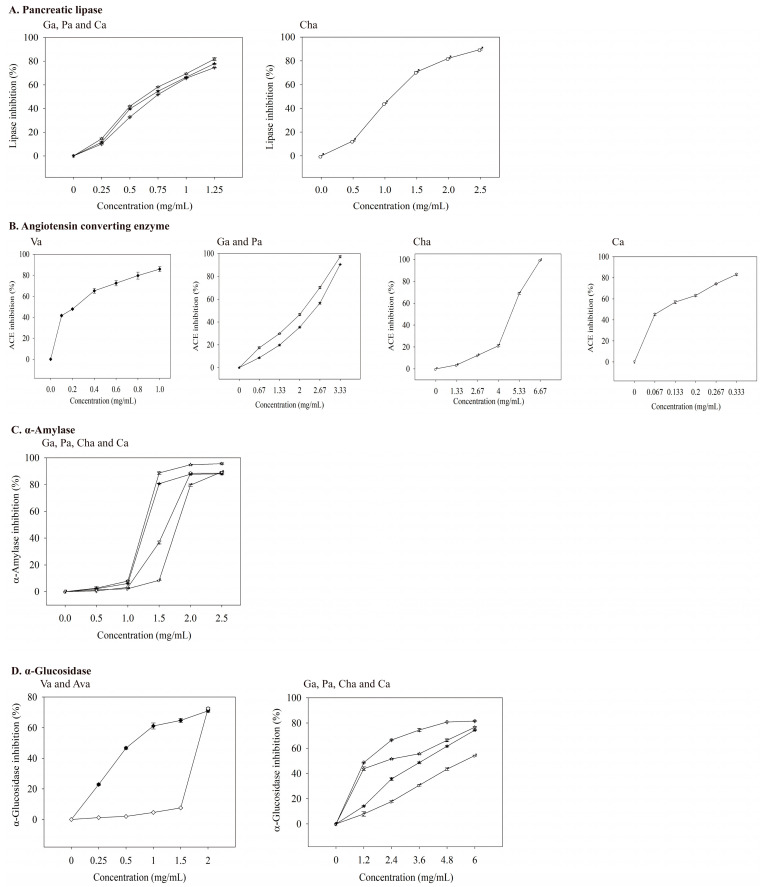
Inhibitory effects of valerenic acid derivatives and phenolic acids against enzymes linked with metabolic syndrome. Each value is expressed as mean ± standard deviation (*n* = 4). Va: Valerenic acid (◆), Ava: Acetoxyvalerenic acid (◊), Ga: Gallic acid (☆), Pa: Protocatechuic acid (★), Cha: Chlorogenic acid (♂), and Ca: Caffeic acid (♀).

**Table 1 foods-12-00636-t001:** Total phenols contents of valerian extracts from roots and rhizomes.

Extraction Method	Roots			Rhizomes		
	Yield (%) ^1^	Total Phenols (mg GAE ^2^/g Sample)		Yield (%)	Total Phenols (mg GAE/g Sample)	
		In Extract	In Powder		In Extract	In Powder
25 °C extract						
25C0E	29.36 ± 0.21C ^3^	14.54 ± 0.53I	4.27 ± 0.18G	29.07 ± 0.15B	13.81 ± 0.33I	4.01 ± 0.09H
25C25E	28.73 ± 0.28D	17.91 ± 0.57G	5.15 ± 0.15E	28.28 ± 0.31C	15.39 ± 0.16H	4.35 ± 0.05F
25C50E	25.20 ± 0.10F	21.12 ± 0.58E	5.32 ± 0.17DE	25.92 ± 0.51E	16.45 ± 0.54G	4.26 ± 0.12FG
25C75E	22.91 ± 0.51H	25.97 ± 0.49C	5.95 ± 0.03C	24.10 ± 0.35G	20.73 ± 0.54D	5.00 ± 0.14C
25C95E	10.24 ± 0.17K	30.31 ± 0.75B	3.10 ± 0.12H	11.47 ± 0.19J	24.69 ± 0.53B	2.83 ± 0.06I
50 °C extract						
50C0E	30.68 ± 0.28A	15.77 ± 0.50H	4.84 ± 0.11F	29.07 ± 0.18B	14.14 ± 0.41I	4.11 ± 0.14GH
50C25E	29.64 ± 0.19C	18.25 ± 0.38G	5.41 ± 0.13D	28.28 ± 0.44C	16.09 ± 0.14GH	4.55 ± 0.06E
50C50E	27.92 ± 0.47E	21.68 ± 0.58E	6.05 ± 0.24C	26.18 ± 0.17E	17.85 ± 0.33F	4.67 ± 0.10DE
50C75E	23.12 ± 0.32H	26.60 ± 0.53C	6.15 ± 0.20C	24.42 ± 0.21FG	22.06 ± 0.69C	5.39 ± 0.21AB
50C95E	13.57 ± 0.22J	32.76 ± 0.92A	4.45 ± 0.06G	15.52 ± 0.25I	25.42 ± 0.75B	3.95 ± 0.17H
75 °C extract						
75C0E	31.07 ± 0.15A	19.33 ± 0.59F	6.00 ± 0.16C	31.37 ± 0.66A	15.44 ± 0.60H	4.84 ± 0.10CD
75C25E	30.18 ± 0.43B	23.97 ± 0.57D	7.23 ± 0.19B	28.76 ± 0.23BC	17.36 ± 0.52F	4.99 ± 0.19C
75C50E	28.21 ± 0.08E	26.80 ± 0.43C	7.56 ± 0.12A	27.50 ± 0.69D	18.97 ± 0.65E	5.22 ± 0.04B
75C75E	24.07 ± 0.23G	29.49 ± 0.70B	7.10 ± 0.16B	25.02 ± 0.46F	22.20 ± 0.71C	5.55 ± 0.12A
75C95E	15.80 ± 0.17I	33.16 ± 0.65A	5.24 ± 0.15DE	17.02 ± 0.11H	26.78 ± 0.89A	4.56 ± 0.13E

^1^ Yield (%) = (freeze-dried extract weight/sample weight) × 100%. ^2^ GAE: Gallic acid equivalent. ^3^ Each value is expressed as mean ± standard deviation (*n* = 4). Means with different capital letters within a column differ significantly (*p* < 0.05).

**Table 2 foods-12-00636-t002:** Phenolic acids contents of valerian extracts.

Extraction Method	Contents (mg/g Lyophilized Extract)					
	Gallic Acid	Protocatechuic Acid	Chlorogenic Acid	Caffeic Acid	Rosmarinic Acid	Total
Roots						
75C0E	0.31 ± 0.01D ^1^	0.57 ± 0.01D	0.21 ± 0.01E	0.57 ± 0.01E	nd ^2^	1.66 ± 0.01E
75C25E	0.33 ± 0.01CD	0.79 ± 0.02C	0.29 ± 0.01D	0.60 ± 0.02D	nd	2.01 ± 0.04D
75C50E	0.35 ± 0.01C	0.82 ± 0.03C	0.30 ± 0.01C	0.64 ± 0.03C	nd	2.11 ± 0.05C
75C75E	0.46 ± 0.01B	0.95 ± 0.03B	0.34 ± 0.01B	0.84 ± 0.04B	nd	2.59 ± 0.05B
75C95E	0.55 ± 0.02A	1.20 ± 0.04A	0.36 ± 0.01A	0.98 ± 0.03A	nd	3.09 ± 0.07A
Rhizomes						
75C0E	0.30 ± 0.01D	0.46 ± 0.02E	0.09 ± <0.01E	0.23 ± <0.01D	nd	1.08 ± 0.03E
75C25E	0.33 ± 0.01C	0.54 ± 0.01D	0.12 ± <0.01D	0.25 ± 0.01D	nd	1.24 ± 0.03D
75C50E	0.39 ± 0.01B	0.61 ± 0.03C	0.13 ± 0.01C	0.28 ± 0.01C	nd	1.41 ± 0.06C
75C75E	0.41 ± 0.01B	0.66 ± 0.03B	0.14 ± <0.01B	0.37 ± 0.02B	nd	1.58 ± 0.06B
75C95E	0.59 ± 0.02A	0.91 ± 0.03A	0.15 ± 0.01A	0.49 ± 0.02A	nd	2.14 ± 0.07A

^1^ Each value is expressed as mean ± standard deviation (*n* = 4). Means with different capital letters within a column differ significantly (*p* < 0.05). ^2^ nd = not detected.

**Table 3 foods-12-00636-t003:** Valerenic acid derivatives contents of valerian extracts.

Extraction Method	Contents (mg/g Lyophilized Extract)							
	Roots				Rhizomes			
	Hva ^1^	Ava	Va	Total	Hva	Ava	Va	Total
25 °C extract								
25C0E	0.34 ± 0.01G ^2^	0.89 ± 0.04K	0.04 ± <0.01K	1.27 ± 0.04N	0.85 ± 0.03H	2.62 ± 0.02K	0.10 ± <0.01J	3.57 ± 0.03M
25C25E	0.46 ± 0.01F	2.23 ± 0.09J	0.13 ± <0.01IJ	2.82 ± 0.10L	0.96 ± 0.03H	5.99 ± 0.26J	0.20 ± 0.01J	7.15 ± 0.28L
25C50E	0.52 ± 0.01F	3.79 ± 0.06H	1.13 ± 0.04G	5.44 ± 0.07I	1.84 ± 0.05F	12.22 ± 0.13G	2.70 ± 0.05G	16.76 ± 0.18I
25C75E	0.80 ± 0.03D	5.07 ± 0.19E	1.44 ± 0.04F	7.31 ± 0.21F	2.60 ± 0.12D	13.89 ± 0.58EF	3.40 ± 0.15F	19.89 ± 0.71F
25C95E	2.77 ± 0.11C	12.65 ± 0.44C	3.34 ± 0.12C	18.76 ± 0.51C	3.84 ± 0.12B	21.81 ± 0.56C	4.35 ± 0.01C	30.00 ± 0.55C
50 °C extract								
50C0E	0.37 ± 0.02G	1.15 ± 0.04K	0.09 ± <0.01JK	1.61 ±0.05MN	0.85 ± 0.03H	3.10 ± 0.11K	0.14 ± <0.01J	4.09 ± 0.12M
50C25E	0.50 ± 0.02F	2.62 ± 0.04I	0.20 ± 0.01I	3.32 ± 0.03K	1.23 ± 0.05G	8.16 ± 0.16I	0.38 ± 0.01I	9.77 ± 0.15K
50C50E	0.63 ± 0.02E	4.27 ± 0.13G	1.47 ± 0.06F	6.37 ± 0.17H	2.00 ± 0.09E	12.26 ± 0.54G	3.36 ± 0.13F	17.62 ± 0.62H
50C75E	0.83 ± 0.03D	5.22 ± 0.11E	1.82 ± 0.03E	7.87 ± 0.12E	2.89 ± 0.13C	14.49 ± 0.41DE	3.83 ± 0.14E	21.21 ± 0.64E
50C95E	3.35 ± 0.08B	13.05 ± 0.28B	3.87 ± 0.09B	20.27 ± 0.36B	4.11 ± 0.19A	23.41 ± 1.05B	4.61 ± 0.10B	32.13 ± 1.03B
75 °C extract								
75C0E	0.48 ± 0.01F	1.17 ± 0.03K	0.09 ± <0.01JK	1.74 ± 0.03M	0.89 ± 0.03H	3.16 ± 0.09K	0.20 ± 0.01J	4.25 ± 0.08M
75C25E	0.52 ± 0.02F	2.90 ± 0.11I	0.30 ± 0.01H	3.72 ± 0.12J	1.15 ± 0.04G	9.25 ± 0.41H	0.64 ± 0.03H	11.04 ± 0.40J
75C50E	0.64 ± 0.01E	4.60 ± 0.06F	1.51 ± 0.02F	6.75 ± 0.07G	2.00 ± 0.09E	13.56 ± 0.44F	3.40 ± 0.14F	18.96 ± 0.63G
75C75E	0.87 ± 0.01D	6.07 ± 0.13D	1.99 ± 0.03D	8.93 ± 0.14D	2.92 ± 0.11C	15.18 ± 0.62D	3.98 ± 0.16D	21.99 ± 0.59D
75C95E	3.48 ± 0.05A	14.26 ± 0.39A	4.16 ± 0.11A	21.90 ± 0.44A	4.25 ± 0.20A	24.20 ± 1.184A	4.77 ± 0.21A	33.22 ± 1.22A

^1^ Hva: Hydroxyvalerenic acid; Ava: Acetoxyvalerenic acid; Va: Valerenic acid; Total = Hva + Ava + Va. ^2^ Each value is expressed as mean ± standard deviation (*n* = 4). Means with different capital letters within a column differ significantly (*p* < 0.05).

**Table 4 foods-12-00636-t004:** EC_50_ values of valerian extracts for scavenging ability on DPPH radicals.

Extraction Method	EC_50_ Value (mg Extract/mL) ^1^	
	Roots	Rhizomes
75C0E	0.708 ± 0.007Ab ^2^	0.928 ± 0.004Aa
75C25E	0.639 ± 0.008Bb	0.853 ± 0.016Ba
75C50E	0.535 ± 0.010Cb	0.765 ± 0.015Ca
75C75E	0.374 ± 0.003Db	0.653 ± 0.007Da
75C95E	0.352 ± 0.004Eb	0.523 ± 0.011Ea

^1^ EC_50_ value: The effective concentration at which DPPH radicals were scavenged by 50%. EC_50_ value was obtained by interpolation from linear regression analysis. EC_50_ values of ascorbic acid, BHA, and α-Tocopherol were 15.67 ± 0.01, 16.48 ± 0.02, and 16.09 ± 0.02 µg/mL, respectively. ^2^ Each value is expressed as mean ± standard deviation (*n* = 4). Means with different capital letters within a column differ significantly (*p* < 0.05). Means with different lowercase letters within a row differ significantly (*p* < 0.05).

**Table 5 foods-12-00636-t005:** IC_50_ values of inhibitory effects of valerian extracts against pancreatic lipase, angiotensin converting enzyme, α-amylase, and α-glucosidase.

Extraction Method	IC_50_ Values (mg Extract/mL) ^1^							
	Pancreatic Lipase		ACE ^2^		α-Amylase		α-Glucosidase	
	Roots	Rhizomes	Roots	Rhizomes	Roots	Rhizomes	Roots	Rhizomes
75C0E	50.75 ± 0.56A ^3^	88.05 ± 1.69A	13.66 ± 0.52A	14.11 ± 0.43A	32.32 ± 0.54A	42.36 ± 0.24A	26.94 ± 0.58A	21.80 ± 2.05A
75C25E	41.84 ± 0.74B	82.38 ± 1.00B	11.22 ± 0.13B	12.52 ± 0.16B	30.41 ± 0.59B	35.47 ± 0.15B	25.92 ± 0.40B	19.28 ± 0.46B
75C50E	38.28 ± 1.40C	47.34 ± 0.99C	4.24 ± 0.10C	4.71 ± 0.06C	28.23 ± 0.48C	29.19 ± 0.86C	18.46 ± 0.69C	18.86 ± 0.13B
75C75E	30.67 ± 0.23D	44.18 ± 0.25D	4.02 ± 0.02C	4.86 ± 0.10C	20.65 ± 0.39D	25.16 ±0.49D	17.99 ± 0.44C	18.59 ± 0.21B
75C95E	17.59 ± 0.82E	40.92 ± 0.54E	3.75 ± 0.09C	4.87 ± 0.10C	12.53 ± 0.10E	21.99 ± 0.27E	15.40 ± 0.35D	17.10 ± 1.02C

^1^ IC_50_ value: The enzyme activity was inhibited by 50%. The IC_50_ value was obtained by interpolation from linear regression analysis. ^2^ ACE: Angiotensin converting enzyme. ^3^ Each value is expressed as mean ± standard deviation (*n* = 4). Means with different capital letters within a column differ significantly (*p* < 0.05).

**Table 6 foods-12-00636-t006:** IC_50_ values of inhibitory effects of orlistat, captopril, acarbose, valerenic acid derivatives and phenolic acids against pancreatic lipase, angiotensin converting enzyme, α-amylase, and α-glucosidase.

	IC_50_ (mg/mL) ^1^			
	Pancreatic Lipase	ACE ^2^	α-Amylase	α-Glucosidase
Orlistat	(9.648 ± 0.032) × 10^−6^E			
Captopril		(0.498 ± 0.014) × 10^−6^F		
Acarbose			(5.404 ± 0.147) × 10^−3^E	(9.547 ± 0.003) × 10^−2^G
Valerenic acid	nd ^3^	0.225 ± 0.023D	nd	0.617 ± 0.028F
Acetoxyvalerenic acid	nd	nd	nd	1.827 ± 0.005D
Gallic acid	0.623 ± 0.002D ^4^	2.100 ± 0.028C	1.258 ± 0.001D	2.164 ± 0.026C
Protocatechuic acid	0.673 ± 0.009C	2.462 ± 0.020B	1.295 ± <0.001C	3.721 ± 0.042B
Chlorogenic acid	1.108 ± 0.002A	4.803 ± 0.004A	1.792 ± 0.003A	5.524 ± 0.074A
Caffeic acid	0.726 ± 0.003B	0.094 ± 0.006E	1.610 ± 0.006B	1.289 ± 0.024E

^1^ IC_50_ value: the enzyme activity was inhibited by 50%. The IC_50_ value was obtained by interpolation from linear regression analysis. ^2^ ACE: Angiotensin converting enzyme. ^3^ nd = not detected. ^4^ Each value is expressed as mean ± standard deviation (*n* = 4). Mean with different capital letters within a column differ significantly (*p* < 0.05).

## Data Availability

Data are contained within the article.

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
