# Peer review of "Bioactive Compounds of Underground Valerian Extracts and Their Effect on Inhibiting Metabolic Syndrome-Related Enzymes Activities"

_foods, 2023, doi:10.3390/foods12030636_

Round 1
Reviewer 1 Report
Interesting manuscript and research work.
Please make some visual corrections in graphs and tables. For instance, there are minor modifications are required in Table 5.
Otherwise, the objectives, methods and the results are clearly presented and discussed.
Introduction:
Please have a look into the literature regarding other advancement in discovering other natural compounds and their effect on inhibition of MS related enzymes.
[Line 42] Authors pointed the “antioxidant activity” of phenolic compounds. That would be essential to add a paragraph regarding the antioxidants, their roll & food nutrients categories (Phenolic compounds are one of these categories).
[Line 56] The paragraph described “MS” should be placed after [Line 49].
Please check the reviews regarding “Herbal therapy in multiple sclerosis”.
It is recommended to ad a few lines regarding the side effects and the risks of taking Valerian in the Introduction.
[Line 71] Was a “24-mesh sieve” sufficient in removing any foreign matter from ingredients (Dried underground valerian parts: the roots and rhizomes)?
[Line 157] Please revise the “edible ethanol” and use the food grade term.
[Line 194-196] As the manuscript discusses Bioactive valerian extracts and their effect on inhibiting MS-related enzymes, the discussion of outcomes requires to be relevant to the subject matter. Please revise across manuscript and focus on the objectives.
[Line 382] Please expand more in conclusion section. The conclusion section currently summarises the outcomes. Please explain the deficiencies of current method, the advantage and disadvantages, other possible methods for extractions rather than solvent extraction.
The rhizomes and roots contained different compounds with different values. What researchers suggest in terms of applicability of these two parts of plant? What are pros and cons using each of these parts, the applied extraction methods and the future applications/solutions?
In manuscript it was stated that “In vitro, the inhibition of R75C95E on pancreatic lipase, α-amylase, α-glucosidase, and ACE was higher than that of Rh75C95E.” Please explain more in detail what are Why are these enzymes inhibitors important?
Author Response
Many thanks to reviewer 1 for the suggestions and comments. Reviewer comments were clearly followed. Please review the revised manuscript as compared to the original manuscript.
Point 1: Please make some visual corrections in graphs and tables. For instance, there are minor modifications are required in Table 5.
Otherwise, the objectives, methods and the results are clearly presented and discussed.
Response 1: The original figure and table are very clear, but they are messed up after uploading to the website. Therefore, the authors have converted all the figures and tables into tiff files (600 dpi) to increase their clarity and visual visibility. Please see p.2, 6-10, 12-14, and 16 of the revised manuscript.
Point 2: Introduction:
Please have a look into the literature regarding other advancement in discovering other natural compounds and their effect on inhibition of MS related enzymes.
Response 2: Thank you for pointing this out. We have added other natural compounds that have an effect on the inhibition of MS-related enzymes in the introduction section. Please see lines 63-66 of the revised manuscript.
Point 3: [Line 42] Authors pointed the “antioxidant activity” of phenolic compounds. That would be essential to add a paragraph regarding the antioxidants, their roll & food nutrients categories (Phenolic compounds are one of these categories).
Response 3: Writing an entire paragraph on the role of antioxidants is beyond the scope of this paper; however, we have added one sentence to explain what antioxidant activity is, which we hope will be useful to the lay reader. Please see lines 42-43 of the revised manuscript.
Point 4: [Line 56] The paragraph described “MS” should be placed after [Line 49].
Response 4: Thanks for your thoughtful suggestion; however, after carefully reviewing all the reviewers’ comments, we decided to keep the presentation as it was.
Point 5: Please check the reviews regarding “Herbal therapy in multiple sclerosis”.
Response 5: We are lost here since we did not mention multiple sclerosis in this paper. Please provide further guidance for us. The MS we addressed in the paper is metabolic syndrome instead of multiple sclerosis.
Point 6: It is recommended to ad a few lines regarding the side effects and the risks of taking Valerian in the Introduction.
Response 6: We have added side effects in the introduction. Please see lines 35-36 of the revised manuscript.
Point 7: [Line 71] Was a “24-mesh sieve” sufficient in removing any foreign matter from ingredients (Dried underground valerian parts: the roots and rhizomes)?
Response 7: The "24 mesh sieve" is sufficient to remove foreign matter from the dried underground valerian parts in this study. Please see the photo in Figure 1 (p.2) of the revised manuscript.
Point 8: [Line 157] Please revise the “edible ethanol” and use the food grade term.
Response 8: The authors have corrected the "edible ethanol" 157 to "food grade ethanol". Please see line 200 of the revised manuscript.
Point 9: [Line 194-196] As the manuscript discusses Bioactive valerian extracts and their effect on inhibiting MS-related enzymes, the discussion of outcomes requires to be relevant to the subject matter. Please revise across manuscript and focus on the objectives.
Response 9: Thanks for reminding us of this. After we have responded to the comments of all four reviewers, the discussion of the results in this paper should be targeted as your comments.
Point 10: [Line 382] Please expand more in conclusion section. The conclusion section currently summarises the outcomes. Please explain the deficiencies of current method, the advantage and disadvantages, other possible methods for extractions rather than solvent extraction.
Response 10: Thank you for brought this up. We have added some points regarding the extraction methods in the conclusion section. Please see lines 446-450 of the revised manuscript.
Point 11: The rhizomes and roots contained different compounds with different values. What researchers suggest in terms of applicability of these two parts of plant? What are pros and cons using each of these parts, the applied extraction methods and the future applications/solutions?
Response 11: Thanks for such a good question that reminds us to consider strategies suitable for commercial or for scientific purposes. As scientists, we would certainly recommend separating the rhizomes from the roots; however, for commercial purposes, separating the two parts of the plant may not be worthwhile as it is time consuming and not cost-effect. If it is acceptable, we prefer to leave this for the readers to imagine.
Point 12: In manuscript it was stated that “In vitro, the inhibition of R75C95E on pancreatic lipase, α-amylase, α-glucosidase, and ACE was higher than that of Rh75C95E.” Please explain more in detail what are Why are these enzymes inhibitors important?
Response 12: We have added an extra sentence to highlight the important point. Please see lines 454-456 of the revised manuscript. (see lines 454-456).

Reviewer 2 Report
This manuscript deals with effects of phenolic and sesquiterpene compounds related to the constituents of roots and rhizomes of Valeriana officinalis on the DPPH radical and on several enzymes including digestive enzymes and ACE. Although this study seems to be valuable for researchers treating quality control of herbal medicines, the following points should be considered by the authors.
1) Although the HPLC conditions of analyzing the extracts are given, no HPLC profiles for the analyses were shown. They should be given in the text.
2) Although some effects of phenolic acids are mentioned in the text, explanations for the effects of valerenic acid and related constituents are poor.
3) Other research groups reported on various types of valerian constituents including sesquiterpenes previously, and therefore the authors should mention the significance of the evaluation of phenolic constituents in the introduction part more clearly.
4) The table forms (especially those of Tables 1, 2, 3, and 5) in the PDF file of the given manuscript have problems to see; actually, quite difficult to read. Check them in the PDF file. They should be improved.
5) Minor problems.
Line 131: “The mobile phase consisted of …” should be “The mobile phase composed of ….”
Lines 194-195: “progression is linked to free radicals scavenging” should be “progression is linked to free radicals.”
Author Response
Many thanks to reviewer 2 for the suggestions and comments. Reviewer comments were clearly followed. Please review the revised manuscript as compared to the original manuscript.
Point 1: Although the HPLC conditions of analyzing the extracts are given, no HPLC profiles for the analyses were shown. They should be given in the text.
Response 1: Since the manuscript already has 4 figures and 6 tables, the authors have included the HPLC profiles of phenolic acids and valerenic acid derivatives in the supplementary information of the revised manuscript (Figure 1S and Figure 2S) for readers' reference. Please see lines 217-219, 223-224, and p.22-23 of the revised manuscript.
Point 2: Although some effects of phenolic acids are mentioned in the text, explanations for the effects of valerenic acid and related constituents are poor.
Response 2: Thanks for bringing this up, which is why we are conducting this study since previous studies have focused on the effects of metabolic syndromes from phenolic acids but not valerenic acid. As we stated in the abstract and text, the current study evidenced valerenic acid has more inhibitory ability of ACE and α-glucosidase than phenolic acids. Thus, we further concluded that additional in vivo studies are needed at the end of this paper.
Point 3: Other research groups reported on various types of valerian constituents including sesquiterpenes previously, and therefore the authors should mention the significance of the evaluation of phenolic constituents in the introduction part more clearly.
Response 3: Thanks for bringing this up, we have added extra information regarding phenolic constituents in the introduction part. Please see lines 63-66 of the revised manuscript.
Point 4: The table forms (especially those of Tables 1, 2, 3, and 5) in the PDF file of the given manuscript have problems to see; actually, quite difficult to read. Check them in the PDF file. They should be improved.
Response 4: The original figure and table are very clear, but they are messed up after uploading to the website. Therefore, the authors have converted all the figures and table files into tiff files (600 dpi) to increase their clarity and visual visibility. Please see p.2, 6-10, 12-14, and 16 of the revised manuscript.
Point 5: Minor problems.
1)Line 131: “The mobile phase consisted of …” should be “The mobile phase composed of ….”
2) Lines 194-195: “progression is linked to free radicals scavenging” should be “progression is linked to free radicals.”
Response 5: We have corrected the minor problems.
1) Please see lines 127 and 136 of the revised manuscript.
2) Please see line 245 of the revised manuscript.

Reviewer 3 Report
We found the manuscript relevant to the ground of this Journal, containing interesting information about Valeriana officinalis extracts while candidates as supplements focusing on metabolic-syndrome issues.
The introduction part covers both old and new references concisely (maybe too briefly) and has a perfect integration of the main aspects of the theme.
The cited core references are relatively recent and appropriate to the discussion. The manuscript is nicely discussed but, in our opinion, a brief allusion to the study’s limitations ought to be done, while aiming at such a complex syndrome.
This article is well written, with a good organization of the contents and a very pertinent methodology, particularly the statistical analysis on extract yields and total phenols, although some suggestions will be done as specific comments.
Regarding enzyme kinetic methodology, and considering the range of this Journal, we found it sufficient, clear, and concisely written ensuring a proper reproducibility of the experience.
The main factor of novelty is the presentation of bioactive Valeriana officinalis extracts that seems a promisor supplement candidate focused on metabolic-syndrome-specific enzyme actors.
Specific comments:
#1_The title (L2-4) must be changed once it induces the reader erroneously to seek the kinetic models used. In this study, no kinetic modeling was attempted. In vitro enzyme activities were accessed, for some metabolic-related enzymes, and the % of inhibition was commonly determined, but no kinetic constants were estimated. Please check thoroughly the manuscript for this misleading issue.
#2_Line 29_for the reasons previously explained, the sentence “….in vitro models, …” must be changed (maybe replacing “models” with “activities”, or removing “in vitro models” from the sentence).
#3_ Figure 1 has good quality, but it seems that the graphs are not of the same quality. Please check if it´s possible to increase the quality of the graphs in Figure 4 (L275).
#4_ Tables 1, 2, and 3 are too dense and difficult to read. We suggest the authors plan a more visually attractive solution, e.g. bar graphs with the letters of statistics above the bars. Another alternative is to select some data and report the full table for supplements. It will be much more interesting for the reader to access the results and compare them.
Author Response
Many thanks to reviewer 3 for the suggestions and comments. Reviewer comments were clearly followed. Please review the revised manuscript as compared to the original manuscript.
Point 1: The title (L2-4) must be changed once it induces the reader erroneously to seek the kinetic models used. In this study, no kinetic modeling was attempted. In vitro enzyme activities were accessed, for some metabolic-related enzymes, and the % of inhibition was commonly determined, but no kinetic constants were estimated. Please check thoroughly the manuscript for this misleading issue.
Response 1: To avoid inducing the reader erroneously to seek the kinetic models used. The authors have corrected the "in vitro models" of the title (L2-4) to "activities". Please see line 4 of the revised manuscript.
Point 2: Line 29_for the reasons previously explained, the sentence “….in vitro models, …” must be changed (maybe replacing “models” with “activities”, or removing “in vitro models” from the sentence).
Response 2: As suggested in point 1, the authors have corrected "in vitro models" to "activities". Please see lines 20, 28-29, 68-69, and 453 of the revised manuscript.
Point 3: Figure 1 has good quality, but it seems that the graphs are not of the same quality. Please check if it´s possible to increase the quality of the graphs in Figure 4 (L275).
Response 3: The original figure and table are very clear, but they are messed up after uploading to the website. Therefore, the authors have converted all the figures and table files into tiff files (600 dpi) to increase their clarity and visual visibility. Please see p.2, 6-10, 12-14, and 16 of the revised manuscript.
Point 4: Tables 1, 2, and 3 are too dense and difficult to read. We suggest the authors plan a more visually attractive solution, e.g. bar graphs with the letters of statistics above the bars. Another alternative is to select some data and report the full table for supplements. It will be much more interesting for the reader to access the results and compare them.
Response 4: The original figure and table are very clear, but they are messed up after uploading to the website. Therefore, the authors have converted all the figures and table files into tiff files (600 dpi) to increase their clarity and visual visibility. Please see p.2, 6-10, 12-14, and 16 of the revised manuscript.

Reviewer 4 Report
The manuscript is easy to read, and most importantly, it compiled a number of procedures that other researchers can easily refer to for similar works. Nevertheless, there are a few issues needed to be addressed. Generally the methodologies need to be added with details and the tables need to be improved for better clarity. The knowledge gap in the study is also not clear. There is no connection between previous works on the plant with the reasons why this study is needed.
1. Line 88, why do you mean by randomly generated?
2. Line 113, what is GAE. need the full name.
3. Line 117 ....volume was adjusted to 5 mL. Why and with what?
4. Please add brief methods for antioxidant properties, Lipase, and amylase assay and etc. Dont just simply cite.
5. The heading for Table 1 is confusing
6. how do you calculate total phenol content and yield.
7. Table 3 data is confusing, please rearrange it.
8. There is no explanation for suddenly focusing only on 75C samples throughout the work.
Author Response
Many thanks to reviewer 4 for the suggestions and comments. Reviewer comments were clearly followed. Please review the revised manuscript as compared to the original manuscript.
Point 1: Line 88, why do you mean by randomly generated?
Response 1: Researchers may carry out quantitative research for different purposes, but two of the important purposes are to establish a causal relationship and to allow the research results to be generalized to a larger scope outside the research context, both of which are closely related to the research design. Randomization should be used to increase the defensibility of the research.
Point 2: Line 113, what is GAE. need the full name.
Response 2: GAE, gallic acid equivalent. The authors have revised the manuscript. Please see lines 118-119 of the revised manuscript.
Point 3: Line 117 ....volume was adjusted to 5 mL. Why and with what?
Response 3: Methanol is volatile and may be lost by ultrasonic vibration. Therefore, use methanol to adjust the sample to 5 mL. The purpose is to facilitate the calculation of the analysis results. The author made a minor correction, please see line 122 of the revised manuscript.
Point 4: Please add brief methods for antioxidant properties, Lipase, and amylase assay and etc. Dont just simply cite.
Response 4: The authors have revised the analytical methods for antioxidant properties and enzyme inhibitory activity. please see lines 144-149, 156-192 of the revised manuscript.
Point 5: The heading for Table 1 is confusing
Response 5: Although the other 3 reviewers did not point out this issue, we should consider all reviewers’ points since the reviewers are advocates for the future readers. Thus, we changed the heading for Table 1 to “Total phenols content of valerian extracts from roots and rhizomes.” Please see p.6 of the revised manuscript.
Point 6: how do you calculate total phenol content and yield.
Response 6:
1) The extraction yield of all treatments is calculated as (dried extract weight/sample weight) × 100%.
2) Total phenols levels of the extracts were analyzed following the method described by Mau et al. [19]. The total phenols content was calculated based on the calibration curve of gallic acid [absorbance at 760 nm = 0.0008 Cgallic acid (mg/mL) + 0.0077, R² = 0.9993]. The results were expressed as milligrams of gallic acid equivalent (GAE) per gram of freeze-dried extract.
3) The authors have revised the manuscript. Please see lines 113-119 of the revised manuscript. There are also notes for Table 1 (p.6).
Point 7: Table 3 data is confusing, please rearrange it.
Response 7: The original figure and table are very clear, but they are messed up after uploading to the website. Therefore, the authors have converted all the figures and table files into tiff files (600 dpi) to increase their clarity and visual visibility. Please see p.2, 6-10, 12-14, and 16 of the revised manuscript.
Point 8: There is no explanation for suddenly focusing only on 75C samples throughout the work.
Response 8: In this study, the 75C samples had the highest content of bioactive components, as we stated in the text. Please see lines 209-230 of the revised manuscript.

Round 2
Reviewer 2 Report
The manuscript was revised adequately, and the explanation for the revision was understandable. Therefore this reviewer recommends publishing this manuscript on this journal.
Reviewer 4 Report
I'm happy with the changes made.